# Essential Oils of Five *Syzygium* Species Growing Wild in Vietnam: Chemical Compositions and Antimicrobial and Mosquito Larvicidal Potentials

**DOI:** 10.3390/molecules28227505

**Published:** 2023-11-09

**Authors:** Le Thi Huong, Nguyen Huy Hung, Nguyen Ngoc Linh, Ty Viet Pham, Do Ngoc Dai, Nguyen Quang Hop, William N. Setzer, Ninh The Son, Wilfried Andlauer, Wolfram Manuel Brück

**Affiliations:** 1School of Natural Science Education, Vinh University, 182 Le Duan, Vinh City 43000, Vietnam; lehuong223@gmail.com; 2Center for Advanced Chemistry, Institute of Research and Development, Duy Tan University, 03 Quang Trung, Da Nang 50000, Vietnam; nguyenhuyhung@duytan.edu.vn; 3Faculty of Pharmacy, Thanh Do University, Kim Chung, Hoai Duc, Hanoi City 10000, Vietnam; 4Faculty of Chemistry, University of Education, Hue University, 34 Le Loi, Hue City 530000, Vietnam; phamvietty@hueuni.edu.vn; 5Faculty of Agriculture, Forestry and Fishery, Nghe An University of Economics, 51-Ly Tu Trong, Vinh City 43000, Vietnam; daidn23@gmail.com; 6Faculty of Chemistry, Hanoi Pedagogical University 2 (HPU2), 32 Nguyen Van Linh, Xuan Hoa, Phuc Yen 15000, Vietnam; nguyenquanghop@hpu2.edu.vn; 7Aromatic Plant Research Center, 230 N 1200 E, Suite 100, Lehi, UT 84043, USA; setzerw@uah.edu; 8Department of Chemistry, University of Alabama in Huntsville, 301 Sparkman, Huntsville, AL 35899, USA; 9Institute of Chemistry, Vietnam Academy of Science and Technology (VAST), 18 Hoang Quoc Viet, Caugiay, Hanoi 10000, Vietnam; 10Institute of Life Technologies, University of Applied Sciences and Arts Western Switzerland Valais, Rue de l’Industrie 19, 1950 Sion, Switzerland; wilfried.andlauer@hevs.ch

**Keywords:** Myrtaceae, *Syzygium*, essential oil, antimicrobial, mosquito larvicidal

## Abstract

The essential oils of five Vietnamese *Syzygium* species (*Syzygium levinei*, *S. acuminatissimum*, *S. vestitum*, *S. cumini*, and *S. buxifolium*) were first hydro-distilled and analyzed using GC-FID/MS (gas chromatography–flame ionization detection/mass spectrometry). Monoterpene hydrocarbons, sesquiterpene hydrocarbons, and oxygenated sesquiterpenoids were the main chemical classes in these oils. All these essential oils showed good–excellent antimicrobial activities against Gram-positive bacteria *Enterococcus faecalis*, *Staphylococcus aureus*, and *Bacillus cereus*, and the yeast *Candida albicans. S. levinei* leaf essential oil, rich in bicyclogermacrene (25.3%), (*E*)-*β*-elemene (12.2%), (*E*)-caryophyllene (8.2%), and *β*-selinene (7.4%), as well as *S. acuminatissimum* fruit essential oil containing (*E*)-caryophyllene (14.2%), *α*-pinene (12.1%), caryophyllene oxide (10.9%), *β*-selinene (10.8%), *α*-selinene (8.0%), and *α*-humulene (5.7%), established the same MIC value of 8 µg/mL against *E. faecalis* and *B. cereus*, which were much better than the positive control streptomycin (MIC 128–256 µg/mL). The studied essential oils showed the potential to defend against mosquitoes since they caused the 24 and 48 h LC_50_ values of less than 50 µg/mL against the growth of *Culex quinquefasciatus* and *Aedes aegypti* larvae. Especially, *S. buxifolium* leaf essential oil strongly inhibited *Ae. aegypti* larvae with 24 and 48 h LC_50_ values of 6.73 and 6.73 µg/mL, respectively, and 24 and 48 h LC_90_ values of 13.37 and 10.83 µg/mL, respectively. These findings imply that Vietnamese *Syzygium* essential oils might have potential for use as supplemental antibacterial agents or as “green” alternatives for the control of mosquitoes.

## 1. Introduction

As a recognized global health problem, antimicrobial resistance among prevalent bacterial infections raises healthcare costs, results in treatment failures, and causes deaths [1]. In developing countries, like Vietnam, where the burden of resistance diseases is disproportionate and there is a dearth of information and knowledge on the burden and epidemiology of these diseases, the problem is particularly urgent.

Infections brought on by Gram (–) bacteria have increased globally in recent years, and in many environments, they are frequently more common than Gram (+) infections [2]. Due to the potential for rapid dissemination of resistance mechanisms and the lack of effective treatments, the risk of antimicrobial resistance among Gram (–) bacteria is increasing and is becoming a global issue [2]. In particular, extensively drug-resistant or multidrug-resistant bacteria which are resistant to three or more classes of antimicrobials are emerging.

Southeast Asia’s tropical climate makes Vietnam home to many mosquito-borne illnesses, such as Japanese encephalitis, dengue fever, and Zika [3]. Japanese encephalitis virus has been always thought to be mostly transmitted by *Culex* species, such as *Culex quinquefasciatus* Say (Diptera: Culicidae) while other mosquito genera might also be effective carriers of the virus [4]. All four dengue virus serotypes are hyperendemic in Vietnam, where they frequently cause acute epidemics of both dengue fever and dengue hemorrhagic fever [3]. The primary carrier of the dengue fever virus in Vietnam is the mosquito *Aedes aegypti* (L.) (Diptera: Culicidae) [3,4]. In 2016, the Zika virus, whose main vector of transmission is the *Aedes* mosquito, was first detected in Vietnam [5]. The growing pesticide resistance in mosquitoes *Aedes* and *Culex* is aggravating this issue [3,4,5].

*Syzygium* is one of the largest genera in the family Myrtaceae, which contains about 1200–1800 species [6,7]. Most plants are shrubs and evergreen trees, which are widely distributed in tropical and subtropical regions; some species are grown as ornamental plants, and a few edible fruits that are used for jams and jellies [7]. *S. aromaticum* (L.) Merr. and L. M. Perry, also known as “clove”, is an important economic species, with its unopened flower buds being used as a well-known spice [8]. Various plants of the genus *Syzygium* have a famous role in ethno-medicine. For instance, *S. cumini* (L.) Skeels has been used in the treatments of diarrhea, dysentery, menorrhagia, and ulcers [9]. Traditionally, *S*. *jambos* L. (Alston) has been recommended for the treatment of hemorrhages, syphilis, leprosy, ulcers, wounds, and lung diseases [10].

As a consequence, a lot of attention has been paid to phytochemical studies of *Syzygium* plants, especially in terms of their essential oils. For instance, the main compounds in clove flower bud essential oil collected from Algeria are phenylpropanoid derivatives such as eugenol (78.72%) [11]. The chemical profile of Egyptian *S. aqueum* Alston leaf essential oil was characterized by *α*-selinene (13.85%) and (*E*)-caryophyllene (12.72%) [12]. In the same manner, *Syzygium* plants growing wild in Vietnam are thought to be a rich essential oil resource. The leaf essential oils of *S. hancei* Merr. and L. M. Perry, gathered from Ha Tinh, Vietnam, were reported to contain the major compounds *γ*-guaiene (11.07%) and *β*-caryophyllene (9.10%) [13]. In another example, *S. szemaoense* Merr. and L. M. Perry leaf essential oil (from Nghe An, Vietnam) with *cis*-*β*-elemene (68.0%) showed antifungal activity against the fungus *Candida albicans* with an MIC value of 64 µg/mL [14]. A recent report by Huong et al. also recommended the use of Vietnamese *S. attopeuense* (Gagnep.), Merr. and L. M. Perry, and *S. tonkinense* (Gagnep.), Merr. and L. M. Perry, leaf essential oils as having the natural potential antimicrobial and mosquito larvicidal agents [15].

The current study aims to identify chemical components in the essential oils of five Vietnamese *Syzygium* species, including *Syzygium levinei* (Merrill.), Merrill, *Syzygium acuminatissimum*, DC., *Syzygium vestitum*, Merr. and Perry, *Syzygium cumini*, and *Syzygium buxifolium*, Hook. and Arn. Additionally, these essential oils were screened for antimicrobial activities and mosquito larvicidal activities.

## 2. Results and Discussion

### 2.1. Essential Oil Compositions

The fresh aerial parts (leaf or leaf and fruit in the cases of BL and BF) of five *Syzygium* plants were collected from central Vietnam (Figure 1). They were hydro-distilled using a Clevenger-type apparatus to obtain the essential oils (Table 1). Then, these essential oils were analyzed using GC-FID/MS, and the outcomes are outlined in Table 2 and Appendix A.

Hydro-distillation of *S. levinei* fresh leaves gave a yellow essential oil with 0.16% yield. Through GC-FID/MS analysis, a total of 44 compounds were identified, which accounted for 93.4% of the composition (Appendix A). Sesquiterpene hydrocarbons were found to achieve the highest percentage of 75.2%, followed by oxygenated sesquiterpenes (16.8%). Monoterpene hydrocarbons and diterpene hydrocarbons were not significant, with 0.9 and 0.5%, respectively. Bicyclogermacrene (25.3%), *trans-β*-elemene (12.2%), (*E*)-caryophyllene (8.2%), and *β*-selinene (7.4%) were the principal compounds in this essential oil sample. Some compounds were found in concentrations of greater than 1.0%, such as viridiflorol (3.5%), cubeban-11-ol (3.0%), *δ*-elemene (2.7%), 1-*epi*-cubenol (2.0%), and germacrene B (1.8%). To date, there has been only one report on the phytochemical characterization of *S. levinei* species [16]. The present study complements this report.

The yellow essential oil of *S. acuminatissimum* fresh leaf was produced with the same yield of 0.16%. As shown in Table 2, 44 compounds were identified, representing 95.7% of the composition (Appendix A). Sesquiterpene hydrocarbons (54.5%) and their oxygenated derivatives (31.6%) were two main chemical classes, whereas the remaining chemical classes included oxygenated monoterpenes (9.0%), monoterpene hydrocarbons (0.3%), and non-terpenes (0.3%). The major compounds include caryophyllene oxide (18.9%), (*E*)-caryophyllene (9.9%), *α*-copaene (9.2%), *α*-cadinene (9.1%), and 1,8-cineole (5.3%). Apparently, there is a remarkable difference between the leaf essential oils of *S. levinei* and *S. acuminatissimum*. Major compounds bicyclogermacrene, *cis-β*-elemene, and *β*-selinene in *S. levinei* leaf oil were completely absent or reached insignificant amount in *S. acuminatissimum* leaf essential oil. In contrast, the major compounds caryophyllene oxide, *α*-copaene, *α*-cadinene, and 1,8-cineole were only found in *S. acuminatissimum*.

The gas chromatographic analysis of *S. acuminatissimum* fruit essential oil showed that 50 compounds (96.9% of the composition) were determined (Appendix A). Sesquiterpene hydrocarbons (51.6%) and their oxygenated derivatives (23.5%) were the two main chemical classes. It turns out that monoterpene hydrocarbons (16.6%) were in higher concentration than those in the leaf essential oil by 16.0%. The fruit also contained oxygenated monoterpenes (4.9%) and non-terpenic compounds (0.3%). This essential oil was dominated by (*E*)-caryophyllene (14.2%), *α*-pinene (12.1%), caryophyllene oxide (10.9%), *β*-selinene (10.8%), *α*-selinene (8.0%), and *α*-humulene (5.7%). As can be seen, (*E*)-caryophyllene was more than that in the leaf essential oil by 4.3%, but caryophyllene oxide was reduced by 8.0%. *α*-Pinene was presented in the leaf essential oil as a trace mount, but it was predominant in the fruit essential oil. *α*-Copaene, *α*-cadinene, and 1,8-cineole appeared as the major compounds in the leaf essential oil, but were less abundant in the fruit essential oil. Likewise, *β*-selinene, *α*-selinene, and *α*-humulene in the fruit essential oil were found in higher concentrations than those in the leaf essential oil. Furthermore, various compounds were only found in the leaf essential oil and vice versa. For instance, camphene, sabinene, *β*-pinene, myrcene, *p*-cymene, *o*-guiacol, *α*-terpineol, *n*-hexyl butanoate, selina-5,11-diene, rosifoliol, and *α*-eudesmol were only found in the fruit essential oil. The chemical compositions of essential oils of *S. acuminatissimum* are presented for the first time in this work.

This is the first time that the chemical composition of essential oil of *S. vestitum* fresh leaf has been reported. The leaf essential oil of *S. vestitum* was also obtained in a yellow color with 0.19% yield. A total of 44 identified compounds are listed in Table 2, corresponding to 95.7% of the composition (Appendix A). Similar to the two first samples, sesquiterpene hydrocarbons (65.5%) and oxygenated sesquiterpenes (29.9%) were the main chemical classes, while monoterpene hydrocarbons were found in only 0.3%. As observed in *S. acuminatissimum* fruit essential oil, (*E*)-caryophyllene (9.2%) and *α*-humulene (9.9%) were found in high concentrations in *S. vestitum* leaf essential oil, whereas the other major compounds, (*E*)-nerolidol (18.9%) and δ-cadinene (9.1%), were only detected in this sample. *S. vestitum* leaf essential oil was also characterized by the presence of various compounds higher than 1.0%, such as *trans-β*-elemene (3.7%), bicyclogermacrene (3.6%), *α*-muurolene (3.6%), (*Z*)-*β*-farnesene (2.5%), germacrene D (2.4%), *β*-selinene (2.3%), humulene epoxide II (2.2%), and humulene epoxide I (2.1%).

*S. cumini* (black plum, jamun, jaman, jambul, or jambolan) is one of the best-known species in the genus *Syzygium* and has been the topic of many phytochemical studies [17]. This article is the first report of the essential oil composition of *S. cumini* fresh leaf collected from central Vietnam. Hydro-distillation of its fresh leaves produced a yellow oil in 0.21% yield. In this essential oil, 30 compounds were identified, representing 94.7% of the composition (Appendix A). It is very different from the essential oil samples described previously since monoterpene hydrocarbons (81.4%) were predominant in this essential oil. Other component classes include oxygenated monoterpenoids (7.0%), oxygenated sesquiterpenoids (3.3%), sesquiterpene hydrocarbons (1.8%), and non-terpenic compounds (1.2%). Two isomers, *α*-pinene (50.4%) and *β*-pinene (23.3%), were the principal compounds in this essential oil. In agreement with previous studies, the pinenes accounted for 22.2 and 4.3% in Brazilian leaf essential oil, and 21.09 and 7.33% in Egyptian leaf essential oil, respectively [18,19]. The abundance of *α*-pinene and *β*-pinene would increase the antioxidative and antibacterial actions of *S. cumini* leaf [20]. The major compound of *S. cumini* leaf essential oil, *α*-pinene, exhibited anti-leishmanial activity via immunomodulation in vitro [21]. Hence, the use of *S. cumini* essential oil and its major constituents for drug development is warranted.

*S. buxifolium* (boxleaf eugenia or fish-scale bush), is a flowering plant native to Vietnam, China, Taiwan, and Japan [22]. It is used as a street tree in several southern Chinese cities [22]. A yellow essential oil was obtained from Vietnamese *S. buxifolium* fresh leaves with a yield of 0.15%. Forty compounds were identified, accounting for 92.7% of the composition (Appendix A). *S. buxifolium* leaf essential oil was accompanied by the appearance of monoterpene hydrocarbons (48.5%), oxygenated sesquiterpenoids (24.5%), sesquiterpene hydrocarbons (19.0%), and diterpene hydrocarbons (0.2%). Myrcene (27.2%), (*E*)-*β*-ocimene (15.8%), *α*-eudesmol (5.7%), and *β*-eudesmol (5.5%) are the main compounds in this essential oil. Other compounds, e.g., *γ*-eudesmol (4.2%), *α*-selinene (3.4%), *β*-selinene (3.1%), (*E*)-caryophyllene (3.0%), and *α*-pinene (2.3%), were also present in significant amounts (Table 2). To date, there is only one report on the chemical profile of Chinese *S. buxifolium* leaf essential oil, in which the main compounds (*E*)-caryophyllene (37.6%), *α*-selinene (9.6%), *β*-selinene (9.4%), and *α*-copaene (5.4%) are mentioned [23].

In general, *(E)*-caryophyllene can be seen as the main compound in essential oils of *Syzygium* species collected from central Vietnam. It was a characteristic compound of *S. hancei* leaf essential oil (9.11%), collected from Ha Tinh [13]. This compound accounted for 11.72 and 80.80% in the leaf essential oils of Nghe An-Vietnamese *S. attopeuense* and *S. tonkinense* [15]. Therefore, the present discoveries serve as additional confirmation that a multitude of factors, notably geographic distribution, can yield identical outcomes in chemical compounds.

### 2.2. Antimicrobial Activity

The obtained essential oil samples were tested for their antimicrobial activity against Gram (+) bacteria *(E. faecalis*, *S. aureus*, *B. cereus*) and Gram (–) bacteria *(E. coli*, *P. aeruginosa*, *S. enterica*), and the yeast *C. albicans*. From Table 3, the tested essential oil samples have generally exhibited remarkable activity against Gram (+) bacterial strains, as compared to Gram (–) bacterial strains. Gram (–) bacterial microorganisms have cell wall lipopolysaccharides, which prevent the lipophilic essential oil components from diffusing into the cells, which has been suggested as the reason why Gram (+) bacteria are more susceptible to the inhibitory effects of essential oils than Gram (–) bacterial organisms [24,25]. In addition, all six samples are better than the standard streptomycin against these Gram (+) bacteria. Especially, *S. levinei* leaf essential oil, containing a high amount of bicyclogermacrene (25.3%), had MIC and IC_50_ values of 8 µg/mL and 4 µg/mL, respectively, much better than those of streptomycin (MIC and IC_50_ of 256 µg/mL and 50.34 µg/mL, respectively). Likewise, *S. acuminatissimum* fruit essential oil established the MIC and IC_50_ values of 8 µg/mL and 2.67 µg/mL, respectively, better than those of streptomycin (MIC and IC_50_ of 128 µg/mL and 20.45 µg/mL, respectively) against bacterium *B. cereus*. The relatively high concentrations of *α*-pinene (12.1%), *β*-selinene (10.8%), *α*-selinene (8.0%), and *α*-humulene (5.7%) in *S. acuminatissimum* fruit essential oil may account for its superior activity compared to *S. acuminatissimum* leaf essential oil against the three tested Gram (+) bacteria (Table 3).

The tested samples also successfully controlled the growth of *C. albicans* with the MIC and IC_50_ values of 16–128 µg/mL and 8.23–65.33 µg/mL, respectively. The positive control cycloheximide resulted in MIC and IC_50_ values of 32 µg/mL and 10.46 µg/mL, respectively. The current results are in good agreement with previous reports, demonstrating the high potential of Vietnamese *Syzygium* essential oils in antimicrobial treatments. A recent report by Huong et al. indicated that the leaf essential oils of *S. attopeuense* and *S. tonkinense* exhibited the strongest activity against *E. faecalis* (MIC and IC_50_ of 4.00 μg/mL and 1.69 μg/mL, respectively) and *C. albicans* (MIC and IC_50_ of 16.00 μg/mL and 8.67 μg/mL, respectively), respectively [15]. The leaf essential oil of *S. corticosum* caused an inhibition to *B. cereus* with the MIC value of 128 µg/mL [14]. Another example is that the leaf essential oil of *S. samarangense* showed very strong activity against *E. coli* with an inhibitory zone of 20.2 mm [26].

### 2.3. Mosquito Larvicidal Activity

Mosquito larvicidal activity of essential oils has been previously characterized with strong (LC_50_ ≤ 50 µg/mL), moderate (50 < LC_50_ ≤ 100 µg/mL), weak (100 < LC_50_ ≤ 750 µg/mL), and inactive (LC_50_ > 750 µg/mL) activity [27]. The 24 h and 48 h larvicidal activities of five studied *Syzygium* species against *Culex quinquefasciatus* and *Aedes aegypti* are summarized in Table 4.

Almost all of the studied essential oils established good mosquito larvicidal activity with the 24 h and 48 h LC_50_ values of less than 50 µg/mL. The best 24 h larvicidal activity was shown for the leaf essential oils of *S. acuminatissimum* and *S. vestitum* against *Cx. quinquefasciatus* (LC_50_ = 16.09 µg/mL), *S. buxifolium* leaf essential oil against *Cx. quinquefasciatus* (LC_50_ = 17.91 µg/mL), and *S. buxifolium* leaf essential oil against *Ae. aegypti* (LC_50_ = 7.67 µg/mL). Correspondingly, the best 24 h LC_90_ values were found in the cases of the leaf essential oils of *S. acuminatissimum*, *S. vestitum*, and *S. buxifolium* against *Cx. quinquefasciatus* and *S. buxifolium* leaf essential oil against *Ae. aegypti* (Table 4).

In the 48 h treatment, the leaf essential oils of *S. acuminatissimum*, *S. vestitum*, and *S. buxifolium* inhibited *Cx. quinquefasciatus*, possessing the best LC_50_ values of 15.34–15.95 µg/mL and LC_90_ values of 19.58–22.72 µg/mL. *S. buxifolium* leaf essential oil exhibited excellent activity against *Ae. aegypti* with LC_50_ value of 6.73 µg/mL and LC_90_ value of 10.83 µg/mL. The leaf essential oils of *S. levinei* and *S cumini* were also noteworthy with an LC_50_ value of about 16 µg/mL.

This is the first time that these *Syzygium* essential oils have been evaluated in mosquito larvicidal assays. A growing body of evidence suggests that *Syzygium* essential oils hold significant potential in larvicidal activity. The leaf essential oils of two other *Syzygium* plants *S. attopeuense* and *S. tonkinense* have demonstrated inhibitory effects against *Ae. aegypti* with the LC_50_ values of 25.55–30.18 μg/mL and LC_90_ values of 33.0–39.01 μg/mL [15]. *Ae. aegypti* larvae (LC_50_ = 92.56 mg/L) were more susceptible to *S. aromaticum* bud essential oil than *Cx. quinquefasciatus* (LC_50_ = 124.42 mg/L) [28]. *S. lanceolatum* leaf essential oil successfully controlled the larvae of *Anopheles stephensi*, *An. subpictus*, *Ae. aegypti*, *Ae. albopictus*, *Cx. quinquefasciatus*, and *Cx. tritaeniorhynchus* with LC_50_ values of 51.2–72.24 μg/mL [29]. *S. zeylanicum* leaf essential oil caused a toxic effect against 3rd instar larvae of *An. subpictus*, *Ae. albopictus*, and *Cx. tritaeniorhynchus*, with LC_50_ values of 83.11, 90.45, and 97.96 μg/mL, respectively [30].

## 3. Materials and Methods

### 3.1. Plant Materials

*Syzygium* plants were gathered from several places in Nghe An and Thanh Hoa provinces of Vietnam (Table 1 and Figure 1). The plant materials were identified by co-author Dr. Do Ngoc Dai, and their voucher specimens were deposited in Nghe An College of Economics. The fresh leaves or fruits (1.5 kg, each) were immediately cut into pieces and hydro-distilled using a Clevenger apparatus for 2.5 h, to obtain the essential oils.

### 3.2. GC-FID/MS Analysis

The essential oils were analyzed using GC-FID as previously reported [31,32]: HP-5 MS column Agilent Technologies (30 m × 0.25 mm, film thickness 0.25 µm), helium carrier gas (1.0 mL/min), injector temperature of 250 °C, and detector temperature of 260 °C. Column temperature program was as follows: 65 °C (4 min hold), increase to 230 °C (4.5 °C/min), 230 °C (9 min hold). We used an inlet pressure of 7.0 kPa, split mode injection (split ratio, 9:1), and injection volume (1.1 µL).

GC-MS measurement was carried out using the same conditions as those used above for GC-FID: Agilent HP 7890A Plus Chromatograph, HP-5 MS column, Agilent HP 5973 MSD mass detector, MS ionization voltage of 70 eV, emission current of 40 mA, acquisitions range of 40–400 amu, a sampling rate of 1.0 scan/s, same operating conditions as above. Retention indices (RI) were determined based on a homologous series of *n*-alkanes (C_7_–C_30_). Essential oil components were identified by co-injection of pure compounds and/or by comparison of RI values and mass spectral fragmentation patterns with those reported in the MS libraries (NIST 17 and Wiley v. 10). On the basis of GC peak area (FID response) and without correction factors; the relative percentage (%) of each compound was calculated. Each analysis was repeated three times.

### 3.3. Antimicrobial Assay

The screening of *Syzygium* essential oils was performed using the broth microdilution assay, as previously described [31,32]. Briefly, three Gram-positive bacteria (*Bacillus cereus* ATCC14579, *Enterococcus faecalis* ATCC29212, and *Staphylococcus aureus* ATCC25923), three Gram-negative bacteria (*Escherichia coli* ATCC25922, *Pseudomonas aeruginosa* ATCC27853, and *Salmonella enterica* ATCC13076), and a yeast (*Candida albicans* ATCC10231) were used for the assays. The bacterial strains were sub-cultured on tryptic soil agar at 37 °C, while *C. albicans* was cultured on potato dextrose agar at 35 °C. The bacterial inocula were adjusted to 5 × 10^5^ CFU/mL and 2.5 × 10^3^ CFU/mL for *C. albicans.* The *Syzygium* essential oils were dissolved in dimethylsulfoxide (DMSO) and diluted in culture media to provide final test concentrations of 4, 16, 32, 64, 128, 256, and 512 μg/mL. Both DMSO (5%) and inoculated wells without antimicrobial agents were used as negative controls; streptomycin (bacteria) and cycloheximide (yeast) were used as positive controls. The test plates were incubated for 24 h at 37 °C (bacteria) or 35 °C (yeast). Resazurin aqueous solution (0.02%) was introduced to the microplates to assess viability. The MIC (minimum inhibitory concentration) was identified as the lowest concentration that showed no apparent growth, whereas the IC_50_ (median inhibitory concentration) was obtained from the optical density measurement and calculated using Graphpad prism 9.5.1.733. Each assay was carried out in triplicate.

### 3.4. Mosquito Larvicidal Assay

*Syzygium* essential oils were screened for mosquito larvicidal activity against *Culex quinquefasciatus* and *Aedes aegypti* as previously described [4,15]. Briefly, *Cx. quinquefasciatus* and *Ae. aegypti* eggs were obtained from the Institute of Biotechnology, VAST, and maintained at the Duy Tan University, Da Nang, Vietnam. Aliquots of each essential oil were dissolved in DMSO to give 1% stock solutions. Into 300 mL beakers, 20 3rd instar larvae and dilutions of the essential oil stock solutions were added to obtain final concentrations of 6.25, 12.5, 25, 50, and 100 µg/mL. Larval mortality was evaluated after 24 h and again after 48 h of exposure. The tests were carried out at room temperature (25 °C ± 2 °C). A DMSO negative control and a permethrin positive control were also carried out. Each test was carried out in quadruplicate. The LC_50_ and LC_90_ values, along with 95% confidence limits, were determined by log-probit analysis using Minitab^®^ 19.2020.1 (Minitab, LLC, State College, PA, USA, 2020).

## 4. Conclusions

For the first time, the present article describes the chemical compositions of five Vietnamese *Syzygium* leaf essential oils and one fruit essential oil, and their antimicrobial and mosquito larvicidal activities. Sesquiterpene hydrocarbons (51.6–75.2%) and oxygenated sesquiterpenoids (16.8–31.6%) were the main chemical classes of the leaf essential oil constituents of *S. levinei*, *S. acuminatissimum*, and *S. vestitum*. The leaf essential oil of *S. cumini* was dominated by monoterpene hydrocarbons (81.4%), whereas that of *S. buxifolium* was characterized by monoterpene hydrocarbons (48.5%), oxygenated sesquiterpenoids (29.5%), and sesquiterpene hydrocarbons (19.0%). The studied essential oils showed good antimicrobial activity against the Gram (+) bacteria *E. faecalis*, *S. aureus*, and *B. cereus* with the MIC and IC_50_ values of 8–128 µg/mL and 2.67–45.67 µg/mL, respectively; this was with the exception of *S. vestitum* leaf essential oil, which had comparable results to those of the positive control streptomycin (MIC and IC_50_ values of 32–356 µg/mL and 20.45–50.34 µg/mL, respectively). Especially, *S. levinei* leaf essential oil, rich in bicyclogermacrene, *trans*-*β*-elemene, (*E*)-caryophyllene, and *β*-selinene, as well as *S. acuminatissimum* fruit containing (*E*)-caryophyllene, *α*-pinene, caryophyllene oxide, *α*-selinene, *β*-selinene, and *α*-humulene, showed the best MIC value of 8 µg/mL against *E. faecalis* and *B. cereus*, respectively.

In addition, the essential oils under study demonstrated the potential to combat mosquitoes since they demonstrated 24 h and 48 h LC_50_ values of less than 50 µg/mL against the larvae of *Cx. quinquefasciatus* and *Ae. aegypti*. With the high content of myrcene, (*E*)-*β*-ocimene, *S. buxifolium* leaf essential oil strongly inhibited *Ae. aegypti* larvae with the best 24 h and 48 h LC_50_ values of 6.73 and 6.73 µg/mL, respectively, and the 24 h and 48 h LC_90_ values of 13.37 and 10.83 µg/mL, respectively. Collectively, the current discoveries underscore the versatile uses of Vietnamese *Syzygium* essential oils in both antimicrobial and mosquito larvicidal treatments. Currently, it is not clear that the essential oils are safe for human or environmental use; there may be potential toxic or detrimental environmental effects. Future research should be carried out to examine potential adverse effects such as mammalian toxicity and toxicity to non-target organisms. In addition, research on formulations that might enhance or prolong the activities of the essential oils should be carried out. Practical applications such as in vivo and in-field experiments should be carried out to assess the practicality of developing *Syzygium* essential oils for use.

## Figures and Tables

**Figure 1 molecules-28-07505-f001:**
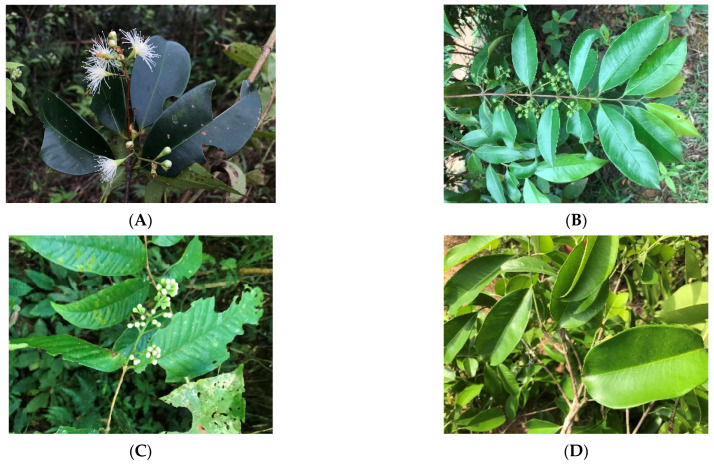
Images of plants examined in this work. (**A**) *Syzygium levinei* (Merrill.) Merrill; (**B**) *Syzygium acuminatissimum* DC; (**C**) *Syzygium vestitum* Merr. and Perry; (**D**) *Syzygium cumini* (L.) Skells; (**E**) *Syzygium buxifolium* Hook. and Arn.

**Table 1 molecules-28-07505-t001:** Plant collection and hydro-distillation details of five Vietnamese *Syzygium* species.

Voucher	Science Name	Local Name	Distribution in Vietnam	Part	Yield (%, *w*/*v*)	Color	Time of Collection	Collection Site
(A) 890L	*Syzygium levinei* (Merrill.) Merrill.	Trâm núi	Thai Nguyen, Phu Tho, Hoa Binh, Thanh Hoa, Nghe An, Ha Tinh, Quang Binh	Leaf	0.16	Yellow	04/2022	Pu Hoat Nature Reserve, Nghean (19°43′43″ N, 104°56′41″ E)
(BL) 894L	*Syzygium acuminatissimum*, DC.	Thoa	Thanh Hoa, Nghe An, Ha Tinh, Kon Tum, Đak Lak	Leaf	0.16	Yellow	05/2022	Pu Huong Nature Reserve, Nghean (19°15′34″ N, 104°55′37″ E)
(BF) 894F	*Syzygium acuminatissimum*, DC.	Thoa	Thanh Hoa, Nghe An, Ha Tinh, Kon Tum, Đak Lak	Fruit	0.12	Yellow	05/2022	Pu Huong Nature Reserve, Nghean (19°15′34″ N, 104°55′37″ E)
(C) 917L	*Syzygium vestitum* Merr. and Perry	Trâm phủ	Lao Cai, Nghe An, Đa Nang	Leaf	0.19	Yellow	04/2022	Pu Hoat Nature Reserve, Nghean (19°42′17″ N, 104°49′46″ E)
(D) 973L	*Syzygium cumini* (L.) Skeels	Vối rừng, Trâm mốc	Lang Son, Vinh Phuc, Ha Noi, Thanh Hoa, Đa Nang, Kon Tum, Lam Đong	Leaf	0.21	Yellow	04/2022	Pu Luong Nature Reserve, Thanhhoa (20°25′21″ N, 105°9′46″ E)
(E) 987L	*Syzygium buxifolium* Hook. and Arn.	Trâm lá cà na	Lao Cai, Nghe An	Leaf	0.15	Yellow	05/2022	Pu Huong Nature Reserve, Nghean (19°18′4″ N, 104°53′59″ E)

**Table 2 molecules-28-07505-t002:** Chemical compositions (%) of essential oils of *Syzygium* species from Vietnam.

RI_calc_	RI_db_	Compounds	A	BL	BF	C	D	E
802	801	Hexanal	---	0.3	---	---	---	---
930	924	*α*-Thujene	---	---	---	---	0.1	---
939	932	*α*-Pinene	---	0.2	**12.1**	---	**50.4**	2.3
953	945	*α*-Fenchene	---	---	---	---	0.1	---
955	946	Camphene	---	---	0.1	---	0.5	---
978	969	Sabinene	---	---	0.3	---	---	---
984	974	*β*-Pinene	---	---	0.6	---	**23.2**	1.8
991	988	Myrcene	---	---	0.3	---	1.5	**27.2**
1005	1004	(*Z*)-Hex-3-enyl acetate	---	---	---	---	1.0	---
1011	1007	Hexyl acetate	---	---	---	---	0.2	---
1029	1022	*p*-Cymene	---	---	0.3	---	1.6	---
1034	1024	Limonene	---	0.1	1.6	0.3	4.0	0.4
1037	1026	1,8-Cineole	---	**5.3**	3.1	---	---	---
1039	1032	(*Z*)-*β*-Ocimene	---	---	---	---	---	0.8
1048	1044	(*E*)-*β*-Ocimene	0.9	---	1.3	---	---	**15.8**
1107	1087	*o*-Guiacol	---	---	0.1	---	---	---
1101	1095	Linalool	---	3.7	1.4	---	0.4	0.3
1121	1128	*allo*-Ocimene	---	---	---	---	---	0.2
1133	1130	*α*-Campholenal	---	---	---	---	0.3	---
1148	1137	*trans*-Sabinol	---	---	---	---	1.2	---
1152	1143	*cis*-Sabinol	---	---	---	---	1.3	---
1172	1160	Pinocarvone	---	---	---	---	0.6	---
1185	1174	Terpinen-4-ol	---	---	---	---	0.2	---
1197	1186	*α*-Terpineol	---	---	0.4	---	0.6	0.2
1191	1191	*n*-Hexyl butanoate	---	---	0.2	---	---	---
1204	1194	Myrtenol	---	---	---	---	0.7	---
1206	1195	Myrtenal	---	---	---	---	0.6	---
1219	1204	Verbenone	---	---	---	---	0.5	---
1294	1284	Bornyl acetate	---	---	---	---	0.2	---
1347	1335	δ-Elemene	2.7	---	---	0.2	---	0.4
1356	1346	*α*-Terpinyl acetate	---	---	---	---	0.4	---
1360	1348	*α*-Cubebene	---	0.7	---	0.1	---	---
1385	1374	Isoledene	0.2	---	---	---	---	---
1388	1374	*α*-Copaene	0.2	**9.2**	2.5	5.3	---	---
1399	1387	*β*-Bourbonene	---	0.5	0.5	---	---	---
1403	1387	*trans*-*β*-Elemene	**12.2**	2.5	0.8	3.7	---	1.6
1424	1411	*cis*-*α*-Bergamotene	---	---	---	0.7	---	---
1425	1413	*α*-Gurjunene	0.5	0.7	0.6	---	---	---
1432	1415	*α*-Cedrene	---	---	---	---	---	0.3
1436	1417	(*E*)-Caryophyllene	**8.2**	**9.9**	**14.2**	**9.2**	0.8	3.0
1444	1431	*β*-Gurjunene	0.2	1.8	0.7	---	---	---
1445	1432	*trans*-*α*-Bergamotene	---	1.3	---	0.5	---	---
1446	1434	γ-Elemene	1.4	---	---	---	---	0.4
1456	1439	Aromadendrene	1.6	0.4	2.4		---	0.9
1459	1440	Selina-5,11-diene	0.1	---	0.3	---	---	---
1460	1440	(*Z*)-*β*-Farnesene	---	---	---	2.5	---	---
1463	1441	*cis*-Muurola-4(14),5-diene	0.2	---	---	---	---	---
1464	1451	*β*-Barbatene	---	---	---	0.7	---	---
1465	1458	*allo*-Aromadendrene	---	---	---	---	---	0.2
1470	1459	α-Humulene	1.3	1.1	**5.7**	**9.9**	0.2	1.0
1475	1461	Striatene	---	---	---	1.8	---	---
1478	1464	9-*epi*-(*E*)-Caryophyllene	2.2	1.1	0.5	1.3	---	0.2
1485	1469	4,5-*di*-*epi*-Aristolochene	---	0.4	0.3	0.4	---	---
1486	1469	Drima-7,9(11)-diene	---	---	---	---	0.2	---
1487	1469	Eudesma-2,4,11-triene	0.8	---	---	---	---	---
1489	1476	*β*-Chamigrene	0.7	2.4	1.2	---	---	0.6
1490	1478	*γ*-Muurolene	---	2.3	---	1.1	0.2	2.1
1491	1483	Bisabola-1,3,5,7(14)-tetraene	---	---	---	1.1	---	---
1493	1483	*α*-Amorphene	1.1	1.2	0.2	0.4	---	---
1496	1485	Germacrene D	2.9	---	---	2.4	---	---
1504	1489	*β*-Selinene	**7.4**	3.6	**10.8**	2.3	0.4	3.1
1508	1493	*trans*-Muurola-4(14),5-diene	---	---	---	1.2	---	---
1511	1494	(*Z*)-*α*-Bisabolene	---	---	---	---	---	0.6
1511	1496	Viridiflorene	0.6	---	---	---	---	---
1512	1498	*α*-Selinene	---	3.6	**8.0**	---	---	3.4
1512	1500	Bicyclogermacrene	**25.3**	---	---	3.6	---	---
1512	1500	*α*-Muurolene	---	---	---	3.6	---	---
1516	1505	*β*-Bisabolene	---	---	---	1.1	---	0.2
1520	1509	*β*-Curcumene	---	---	---	0.1	---	---
1521	1511	*δ*-Amorphene	1.4	---	---	---	---	0.1
1529	1513	*γ*-Cadinene	0.2	1.1	0.7	0.5	---	0.3
1536	1519	*δ*-Cadinene	1.1	0.1	0.7	**9.1**	---	---
1537	1521	*trans*-Calamenene	0.2	0.5	0.5	---	---	---
1540	1528	Zonarene	0.2	---	---	1.0	---	0.2
1546	1533	*trans*-Cadina-1,4-diene	---	---	---	0.5	---	---
1550	1536	(*E*)-*α*-Bisabolene	---	---	---	0.4	---	---
1552	1537	*α*-Cadinene	0.3	**9.1**	0.2	---	---	---
1558	1544	*α*-Calacorene	---	1.0	0.8	0.8	---	---
1559	1546	Selina-3,7(11)-diene	0.2	---	---	---	---	---
1565	1548	Elemol	---	---	---	---	---	2.7
1569	1551	(*E*)-Nerolidol	---	0.5	0.5	**18.9**	0.2	0.3
1576	1559	Germacrene B	1.8	---	---	---	---	0.4
1587	1567	Palustrol	0.7	0.4	0.3	---	---	---
1589	1570	Caryophyllenyl alcohol	---	---	---	0.5	---	---
1595	1577	Spathulenol	2.1	0.8	2.0	0.5	---	2.2
1602	1592	Viridiflorol	3.5	---	---	---	---	---
1603	1593	Caryophyllene oxide	---	**18.9**	**10.9**	1.3	2.7	---
1612	1595	Cubeban-11-ol	3.0	0.5	1.0	---	---	0.3
1613	1600	Guaiol	0.4	---	---	0.5	---	0.2
1618	1600	Humulene epoxide I	---	0.5	0.3	2.1	---	
1620	1600	Rosifoliol	0.4	---	0.3	---	---	---
1621	1601	Curzerenone	---	---	---	---	---	0.5
1623	1602	Ledol	0.5	1.3	0.3	---	---	---
1628	1602	6-*epi*-Cubenol	0.2	---	---	---	---	---
1628	1608	*epi*-Cedrol	---	---	---	---	---	2.6
1629	1608	Humulene epoxide II	---	0.5	2.8	2.2	0.4	---
1644	1627	1-*epi*-Cubenol	2.0	2.2	0.4	0.8	---	---
1641	1637	5-Guaiene-11-ol	---	2.1	0.4	---	---	---
1649	1638	*γ*-Eudesmol	---	0.8	0.3	---	---	4.2
1651	1638	Humulene epoxide III	---	0.7	---	---	---	---
1657	1638	*epi*-*α*-Cadinol	1.0	0.5	1.0	0.7	---	0.3
1658	1640	*epi*-*α*-Muurolol	0.5	---	---	---	---	---
1661	1644	*α*-Muurolol	---	---	---	0.5	---	---
1670	1649	*β*-Eudesmol	---	0.8	0.2	---	---	**5.5**
1671	1652	*α*-Cadinol	1.1	---	---	0.8	---	---
1672	1652	*α*-Eudesmol	---	---	0.7	---	---	**5.7**
1674	1658	*neo*-Intermedeol	1.1	0.6	1.9	0.6	---	---
1684	1666	Bulnesol	---	---	---	0.3	---	---
1687	1668	14-Hydroxy-9-*epi*-(*E*)-Caryophyllene	---	0.3	0.2	---	---	---
1695	1683	*epi*-*α*-Bisabolol	---	---	---	0.2	---	---
1770	1710	*α*-Cyperone	---	0.2	---	---	---	---
1812	1790	Eudesma-3,11-dien-2-one	0.3	---	---	---	---	---
2116	1942	Phytol	0.5	---	---	---	---	0.2
		**Total**	**93.4**	**95.7**	**96.9**	**95.7**	**94.7**	**92.7**
		Monoterpene hydrocarbons	0.9	0.3	16.6	0.3	81.4	48.5
		Oxygenated monoterpenes	---	9.0	4.9	---	7.0	0.5
		Sesquiterpene hydrocarbons	75.2	54.5	51.6	65.5	1.8	19.0
		Oxygenated sesquiterpenes	16.8	31.6	23.5	29.9	3.3	24.5
		Diterpene hydrocarbons	0.5	---	---	---	---	0.2
		Non-terpenic compounds	---	0.3	0.3	---	1.2	---

RI_calc_ = retention indices determined with respect to a homologous series of C_7_–C_30_
*n*-alkanes on an HP-5 MS column; RI_db_ = retention indices from the databases (NIST 17 and Wiley version 10); A—*S. levinei* leaf; BL—*S. acuminatissimum* leaf; BF—*S. acuminatissimum* fruit; C—*S. vestitum* leaf; D—*S. cumini* leaf; E—*S. buxifolium* leaf; bold—compounds with greater than 5.0%. Bold number: major components.

**Table 3 molecules-28-07505-t003:** Antimicrobacterial activities of Vietnamese *Syzygium* essential oils.

Species	Gram (+) Bacteria			Gram (–) Bacteria			Yeast
MIC (µg/mL)							
	*E. faecalis*	*S. aureus*	*B. cereus*	*E. coli*	*P. aerguginosa*	*S. enterica*	*C. albicans*
*S. levinei* leaf	8	64	32	-	32	-	16
*S. acuminatissimum* leaf	64	128	32	-	32	-	16
*S. acuminatissimum* fruit	32	32	8	-	32	-	128
*S. vestitum* leaf	128	128	256	-	-	-	64
*S cumini* leaf	16	32	16	128	-	-	128
*S. buxifolium* leaf	32	64	64	-	-	-	128
Streptomycin	256	256	128	32	256	128	
Cycloheximide							32
IC_50_ (µg/mL)							
*S. levinei* leaf	4.00	17.67	12.45	-	9.67	-	8.23
*S. acuminatissimum* leaf	31.68	45.67	9.78	-	11.23	-	8.33
*S. acuminatissimum* fruit	17.00	10.45	2.67	-	9.67	-	65.33
*S. vestitum* leaf	43.23	37.34	100.89	-	-	-	23.56
*S cumini* leaf	10.34	20.34	34.78	-	-	-	56.78
*S. buxifolium* leaf	4.98	3.78	5.78	-	-	-	28.79
Streptomycin	50.34	45.24	20.45	9.45	41.46	45.67	
Cycloheximide							10.46

MIC—minimum inhibitory concentration; IC_50_—median inhibitory concentration; “-“—inactive.

**Table 4 molecules-28-07505-t004:** Mosquito larvicidal activities of *Syzygium* essential oils from Vietnam.

Essential Oils	LC_50_ (95% Limits)	LC_90_ (95% Limits)	χ^2^	*p*
	24 h treatment			
	*Culex quinquefasciatus* (3rd instar)			
*S. levinei* leaf	38.94 (35.64–42.00)	52.37 (48.28–58.37)	0.0985	0.999
*S. acuminatissimum* leaf	16.09 (14.88–17.52)	22.79 (20.50–26.56)	0.3168	0.989
*S. acuminatissimum* fruit	34.15 (31.09–37.53)	59.10 (52.07–70.29)	2.2694	0.686
*S. vestitum* leaf	16.09 (14.88–17.52)	22.79 (20.50–26.56)	0.3168	0.989
*S cumini* leaf	50.85 (46.90–55.20)	78.88 (70.28–94.12)	7.7500	0.051
*S. buxifolium* leaf	17.91 (16.46–19.45)	23.93 (21.84–27.02)	0.0642	0.999
Permethrin (control)	0.0094 (0.0082–0.0107)	0.0211 (0.0185–0.0249)	57.6	0.000
	*Aedes aegypti*			
*S. levinei* leaf	20.70 (18.74–22.79)	37.34 (32.81–44.51)	6.1343	0.189
*S. acuminatissimum* leaf	24.20 (22.60–25.70)	32.61 (29.77–39.23)	0.6686	0.955
*S. acuminatissimum* fruit	24.04 (22.42–25.52)	32.37 (29.59–38.81)	0.6069	0.962
*S. vestitum* leaf	24.20 (22.60–25.70)	32.61 (29.77–39.23)	0.6686	0.955
*S cumini* leaf	20.97 (19.15–23.00)	36.94 (32.59–43.73)	5.8406	0.120
*S. buxifolium* leaf	7.67 (6.98–8.43)	13.37 (11.76–15.97)	1.3290	0.856
	48 h treatment			
	*Culex quinquefasciatus* (3rd instar)			
*S. levinei* leaf	33.50 (30.92–36.46)	47.20 (42.62–54.49)	0.2492	0.993
*S. acuminatissimum* leaf	15.95 (14.75–17.37)	22.72 (20.42–26.53)	0.3679	0.985
*S. acuminatissimum* fruit	31.85 (28.91–35.13)	57.82 (50.58–69.41)	5.3403	0.254
*S. vestitum* leaf	15.95 (14.75–17.37)	22.72 (20.42–26.53)	0.3679	0.985
*S cumini* leaf	44.00 (40.11–48.38)	79.25 (69.51–94.68)	9.6975	0.021
*S. buxifolium* leaf	15.34 (14.28–17.05)	19.58 (17.49–24.07)	0.0278	1.000
	*Aedes aegypti*			
*S. levinei* leaf	16.77 (15.22–18.48)	30.24 (26.53–36.08)	2.5974	0.627
*S. acuminatissimum* leaf	22.77 (21.06–24.33)	31.79 (29.11–36.77)	0.6796	0.954
*S. acuminatissimum* fruit	23.60 (21.91–25.23)	33.22 (30.25–38.94)	1.1100	0.893
*S. vestitum* leaf	22.77 (21.06–24.33)	31.79 (29.11–36.77)	0.6796	0.954
*S. cumini* leaf	16.15 (14.76–17.70)	26.80 (23.74–31.73)	3.3271	0.344
*S. buxifolium* leaf	6.73 (6.18–7.34)	10.83 (9.61–12.91)	0.5554	0.968

LC_50_—50% lethal concentration; LC_90_—90% lethal concentration; χ^2^ and *p*—goodness-of-fit chi-square value and *p*-value, respectively.

## Data Availability

Data are contained within the article.

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
