# Peer review of "Essential Oils of Five Syzygium Species Growing Wild in Vietnam: Chemical Compositions and Antimicrobial and Mosquito Larvicidal Potentials"

_molecules, 2023, doi:10.3390/molecules28227505_

Round 1
Reviewer 1 Report
Comments and Suggestions for Authors
Dear authors
Your manuscript is well-designed. the fact that there are not many studies in the literature on the subject of this manuscript makes your research unique. looking at the manuscript in general, I think that the evaluation of the discussion part by comparing it with a few more different literature studies will make it easier for the readers to understand the originality of this manuscript.
For example, the following studies may help you with this discussion section
1-A Detailed Biological and Chemical Investigation of Sixteen Achillea Species’ Essential Oils via Chemometric Approach
2-Essential oil content, in-vitro and in-silico activities of Hypericum triquetrifolium Turra, H. empetrifolium subsp. empetrifolium Willd., and H. pruinatum Boiss. & Balansa species
Author Response
Reviewer 1
Your manuscript is well-designed. the fact that there are not many studies in the literature on the subject of this manuscript makes your research unique. looking at the manuscript in general, I think that the evaluation of the discussion part by comparing it with a few more different literature studies will make it easier for the readers to understand the originality of this manuscript.
For example, the following studies may help you with this discussion section
1-A Detailed Biological and Chemical Investigation of Sixteen Achillea Species’ Essential Oils via Chemometric Approach
2-Essential oil content, in-vitro and in-silico activities of Hypericum triquetrifolium Turra, H. empetrifolium subsp. empetrifolium Willd., and H. pruinatum Boiss. & Balansa species
Response: The authors thank you for your suggestion. However, these references seem to not relate this submission. Therefore, we would like to retain without citing these references
Reviewer 2 Report
Comments and Suggestions for Authors
The manuscript “Essential oils of five Syzygium species growing wild in Vietnam: Chemical compositions, antimicrobial and mosquito larvicidal potentials” reports interesting and novel results on the antimicrobial and mosquitocidal activity of different essential oil obtained from plants of the genus Syzygium. However, in the current form, it is not suitable for publications. There are many points that should be revised:
1. In some cases, the same reference has been reported twice in the same sentence. (See Line 63 and 72-73). The authors should cite the reference once in the sentence.
2. Some sentences are grammatically incorrect (See line 72-73, 133-34, 142-143, 147-148). The English language should be revised in the whole manuscript.
3. The authorship of plants, bacteria, and mosquitos should be reported the first time they are cited in the manuscript. For example, S. cumini should be fully written and the authorship should be added in line 76, while it should be abbreviated in lines 94/95.
4. Sometimes the scientific names of plants have not been reported in italics (See lines 76,77). Please, revise all the scientific names in the whole manuscript.
5. The authors reported “oil” when referring to “essential oil”. They are not the same thing. Please correct with “essential oil”. The authors could abbreviate “essential oil” with EO the first time it is reported in the manuscript and abbreviate in the rest of the manuscript.
6. Line 74: what authors mean with “most crucially economic”? The sentence is not clear. Please, rephrase.
7. In line 84-84 the authors stand that the plants of the genus Syzygium are rich in essential oil because of the studies previously reported in lines 81-83. There is not any correlation between the sentences. Please improve this part of the introduction. Moreover, the authors should improve the whole part of the introduction referring to Syzygium genus, which should be presented more clearly.
8. The last sentence of the introduction (Lines 95-97) should also be rephrased.
9. Line 100: The abbreviation of BL and BF should be explained.
10. Table 1: Please, replace “time” with “time of collection”. Moreover, the coordinates of the collection sites should be added.
11. Section 2.1: There are many errors in this section. Please, improve the sentences.
For example, “accounted for 93.4%” is not clear. “93.4 %” of what?
What do you mean by “phytochemical isolation”? “Phytochemical characterization” is more appropriate (Lines 118/119).
Compounds have not been identified from the table, but from the EO sample (Lines 122-123).
Line 137: “Non-terpenes” is incorrect. Please, replace it.
Lines 201-203: this sentence is totally incorrect.
12. The GC-MS analysis should have been performed in duplicate or triplicate to verify the repeatability of the analysis.
13. Section 2.2: The MIC and IC50 are reported in a confusing manner in the text (8 μg/mL/4 μg/mL). Please, separate the two values. Moreover, in the experimental section it has been reported that the assays were repeated in triplicate. Where is the standard deviation of the values?
14. Section 3.2: The first sentence is too long and is not well-written.
15. Section 3.3: How many concentrations of each essential oil did you test to calculate the MIC and the IC50 values? It is not clear. How did you calculate the IC50 values? It was not reported.
16. Section 3.4: How many concentrations of each essential oil did you test to calculate le LC50 values? It is not clear.
17. Line 325: “of the leaf EO constituents”.
18. What about the future perspectives of this study? Please, add some sentences about the limits of the study and potential future developments.
Comments on the Quality of English Language
The quality of English in which the manuscript has been written is somewhat mediocre. It should be reviewed by a native speaker before being considered for publication.
Author Response
Reviewer 2
The manuscript “Essential oils of five Syzygium species growing wild in Vietnam: Chemical compositions, antimicrobial and mosquito larvicidal potentials” reports interesting and novel results on the antimicrobial and mosquitocidal activity of different essential oil obtained from plants of the genus Syzygium. However, in the current form, it is not suitable for publications. There are many points that should be revised:
- In some cases, the same reference has been reported twice in the same sentence. (See Line 63 and 72-73). The authors should cite the reference once in the sentence.
Response: The authors thank you for this point. This is removed. Please see.
- Some sentences are grammatically incorrect (See line 72-73, 133-34, 142-143, 147-148). The English language should be revised in the whole manuscript.
Response: The manuscript has been proof-read by a native speaker of English (co-author: professor William) and these sentences have been corrected.
- The authorship of plants, bacteria, and mosquitos should be reported the first time they are cited in the manuscript. For example, S. cumini should be fully written and the authorship should be added in line 76, while it should be abbreviated in lines 94/95.
Response: It is correct followed your suggestions. Please see
- Sometimes the scientific names of plants have not been reported in italics (See lines 76,77). Please, revise all the scientific names in the whole manuscript.
Response: The italics has been corrected for entire manuscript.
- The authors reported “oil” when referring to “essential oil”. They are not the same thing. Please correct with “essential oil”. The authors could abbreviate “essential oil” with EO the first time it is reported in the manuscript and abbreviate in the rest of the manuscript.
Response: “Essential oil” was applied to entire manuscript. Please see.
- Line 74: what authors mean with “most crucially economic”? The sentence is not clear. Please, rephrase.
Response: The sentence has been corrected.
- In line 84-84 the authors stand that the plants of the genus Syzygium are rich in essential oil because of the studies previously reported in lines 81-83. There is not any correlation between the sentences. Please improve this part of the introduction. Moreover, the authors should improve the whole part of the introduction referring to Syzygium genus, which should be presented more clearly.
Response: In this paragraph, we would like to add two words “for instance” and “In the same manner”. The next sentences describe the studies on Syzygium essential oil in either the world and Vietnam, which explain the idea of the first sentence.
- The last sentence of the introduction (Lines 95-97) should also be rephrased.
Response: The sentence has been corrected.
- Line 100: The abbreviation of BL and BF should be explained.
Response: We would like to remove these words since Table 1 and footnote of Table 2 have indicated the mean of BL and BF. Please see.
- Table 1: Please, replace “time” with “time of collection”. Moreover, the coordinates of the collection sites should be added.
Response: This information was added. Please see.
- Section 2.1: There are many errors in this section. Please, improve the sentences.
For example, “accounted for 93.4%” is not clear. “93.4 %” of what?
Response: 93.4% of the composition. This is correct for six oil samples. Please see
What do you mean by “phytochemical isolation”? “Phytochemical characterization” is more appropriate (Lines 118/119).
Response: It is replaced. Please see
Compounds have not been identified from the table, but from the EO sample (Lines 122-123).
Response: As shown in Table 2, ………….
Line 137: “Non-terpenes” is incorrect. Please, replace it.
Response: Non-terpenic compounds.
Lines 201-203: this sentence is totally incorrect.
Response: This sentence has been modified as “In general, (E)-caryophyllene can be seen as the main compound in essential oils of Syzygium species, collected from Central Vietnam”
- The GC-MS analysis should have been performed in duplicate or triplicate to verify the repeatability of the analysis.
Response: GC-MS was run only one time. However, the accuracy is very high since the result is mostly based on GC-FID.
- Section 2.2: The MIC and IC50 are reported in a confusing manner in the text (8 μg/mL/4 μg/mL). Please, separate the two values. Moreover, in the experimental section it has been reported that the assays were repeated in triplicate. Where is the standard deviation of the values?
Response: We suppose that the MIC/IC50 is equal to the MIC and IC50. We had published various papers in the same manner. Regarding standard deviation, it is around minus and plus 0.001, therefore we don’t provide here.
- Section 3.2: The first sentence is too long and is not well-written.
Response: The sentence has been modified.
- Section 3.3: How many concentrations of each essential oil did you test to calculate the MIC and the IC50 values? It is not clear. How did you calculate the IC50 values? It was not reported.
Response:
-To provide final test concentrations of 4, 16, 32, 64, 128, 256 and 512 μg/mL
-The MIC (minimum inhibitory concentration) was identified as the lowest concentration that showed no apparent growth, whereas the IC50 (median inhibitory concentration) was obtained from the optical density measurement and calculated using Graphpad prism 9.5.1.733
- Section 3.4: How many concentrations of each essential oil did you test to calculate le LC50 values? It is not clear.
Response: Dilutions of the essential oil stock solutions to give final concentrations of 6.25, 12.5, 25, 50 and 100 µg/mL
- Line 325: “of the leaf EO constituents”.
Response: It is added. Please see
- What about the future perspectives of this study? Please, add some sentences about the limits of the study and potential future developments.
Response: This has been added to the Conclusions section.
Reviewer 3 Report
Comments and Suggestions for Authors
-Use the respective italic letters in lines: 76,77, 128.
-Line 88: correct “antifungal”
- Change “Syzygium acuminatissimum (Blume) DC.” by “Syzygium acuminatissimum DC.” [https://www.worldfloraonline.org/]
-Change “Syzygium levinei (Merrill) Merrill & L. M. Perry” by “Syzygium levinei (Merrill.) Merrill.”
-Specify who identified the plant material studied, and which herbarium was used.
-Title of Table 1: …” of the studied species…”
-photo (A): this photo is irrelevant, lacking quality or specificity.
-Search by previous articles published with synonyms of these species, e.g.: doi:10.1016/S0367-326X(02)00131-4; doi.org/10.1080/0972060X.2016.1232608 for S. cumini.
Author Response
Reviewer 3
-Use the respective italic letters in lines: 76,77, 128.
Response: The authors thank you for revision. They are italic now. Please see
-Line 88: correct “antifungal”
Response: It is correct now. Please see
- Change “Syzygium acuminatissimum (Blume) DC.” by “Syzygium acuminatissimum DC.” [https://www.worldfloraonline.org/]
Response: It is correct now. Please see
-Change “Syzygium levinei (Merrill) Merrill & L. M. Perry” by “Syzygium levinei (Merrill.) Merrill.”
Response: It is correct now. Please see
-Specify who identified the plant material studied, and which herbarium was used.
Response: It is correct now. Please see “Plant materials”
-Title of Table 1: …” of the studied species…”
Response: It is correct now. Please see
-photo (A): this photo is irrelevant, lacking quality or specificity.
Response: It is replaced by a new photo. Please see.
-Search by previous articles published with synonyms of these species, e.g.: doi:10.1016/S0367-326X(02)00131-4; doi.org/10.1080/0972060X.2016.1232608 for S. cumini.
Response: Essential oil of S. cumini in the world has been studied, but essential oil of this species collected from Vietnam is documented for the first time.
Round 2
Reviewer 2 Report
Comments and Suggestions for Authors
The authors have incorporated all the suggestions, and the manuscript has improved in quality. For the reasons mentioned, the manuscript can be accepted for publication
Comments on the Quality of English LanguageThe English has been significantly improved as the manuscript has been reviewed by a native speaker